# COVID-19 and Islamic Stock Index: Evidence of Market Behavior and Volatility Persistence

**Adil Saleem** [1] 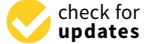**, Judit Bárczi** [1] **and Judit Sági** [2,*]

1  Doctoral School of Economics and Regional Sciences, Hungarian University of Agriculture and Life Sciences, H-2100 Gödöllő, Hungary; saleem.adil@phd.uni-szie.hu (A.S.); barczi.Judit@uni-mate.hu (J.B.)
2  Faculty of Finance and Accountancy, Budapest Business School-University of Applied Sciences, H-1149 Budapest, Hungary
*  Correspondence: sagi.judit@uni-bge.hu

**Abstract:** The aftermath of the COVID-19 pandemic is not limited to human lives and health sectors. It has also changed social and economic aspects of the world. This study investigated the Islamic stock market's reaction and changes in volatility before and during this pandemic. The market model of event study methodology was employed to analyze Islamic stock market reactions in nine different markets around the globe. To examine changes in volatility and persistence of risk, the generalized autoregressive conditional heteroscedasticity (GARCH) method was used. Nine Islamic stock indices were selected for this study from the Thomson Reuters data stream. The results suggest that, in the short run, the Islamic Australian stock index and Islamic GCC stock index remained stable for the first 15 days following news of the pandemic. The Islamic stock indexes of Qatar, UAE, ASEAN, MENA, MENASA, and Bahrain were significantly affected by the outbreak in the short-term. On the other hand, the volatility of Islamic stock indices was substantially amplified after the global health crisis was declared by the WHO. Moreover, volatility shocks tended to persist for a longer period after COVID-19.

**Keywords:** Islamic stocks; COVID-19; abnormal return; volatility

## 1. Introduction

Since the beginning of 2020, the world has been alarmed by a new virus named COVID-19. According to health practitioners, the virus is fatal for up to 2–3% of those infected. According to the Wuhan municipal health committee (WMHC), on 30th December 2019, four patients presenting symptoms similar to pneumonia were admitted to a municipal hospital in Wuhan, and the diagnosis could not be made in time (ProMED 2019; Sohrabi et al. 2020). The cases of this unknown disease increased significantly by mid-January 2020 despite a huge influx of multidisciplinary task force workers, as recommended by WMHC. Keeping in mind the situation in Wuhan, in a press conference conducted by the WHO (2020) in mid-January 2020, the presence of human-to-human spread was disclosed. Furthermore, a special task force (Emergency Committee–EC), based on 15 multinational health experts, was formulated by the WHO to address this issue. Importantly, the WHO (2020) proclaimed COVID-19 a global health emergency as it reached the territories of Germany, Vietnam, the UK, and Japan. Soon after the breakout of COVID-19, several economic and social changes occurred, with immediate effects in China and other parts of the world. In the affected places, economic activities were locked down and social events were limited. Additionally, the virus was found in more than 190 countries by the end of March 2020. Therefore, on 11th March 2020, COVID-19 was announced as a pandemic, affecting millions of people (Dunford et al. 2020). However, it is imperative to note that the long-term economic impact of viral outbreaks is more destructive than

its actual fatality rate (Smith 2006). Similarly, the recent global health crisis triggered by COVID-19 is expected to impart severe economic, social, and financial effects on a global scale (Goodell 2020).

Lee and McKibbin (2004) examined the economic impact of an epidemic outbreak from a virus in the same family as COVID-19, severe acute respiratory syndrome (SARS), in selected countries. They documented a drastic decline in consumption and investment patterns in the corporate sector. As reported in a behavioral finance study by MacKinlay (1997), market news related to risks and contagious disease can make investors passive and ultimately affect investors' sentiments as whole. Conventional stock markets and the history of pandemics have been examined by many authors, including but not limited to: Lee and McKibbin (2004); In et al. (2002); Zhang et al. (2020); Topcu and Gulal (2020); and Liu (2020). Importantly, Donadelli et al. (2017) argued that news-related health issues, especially those sparked by the WHO, have had a detrimental effect on investors' sentiments and affected their decisions. Furthermore, in the past, conventional stock markets experienced major setbacks due to a global health emergency promulgated by the WHO (Wang et al. 2013).

Islamic finance constitutes a substantial part of the global financial and capital market with innovative alternatives. The exponential growth in Islamic finance assets has been remarkable and is still expanding progressively (Paltrinieri et al. 2019; Saleem and Ashfaque 2020). Islamic finance is defined as financing activities based on Islamic principles, which prohibit dealing in interest, speculation, gambling, and uncertain transactions. Apart from permissible business activity, stocks must comply with strict screening criteria in order to be considered in the Islamic index. Thus, a company must have very low or zero interest income, a low debt ratio and small cash holdings (El-Gamal 2000), requirements which are completely ignored in the conventional financial system (Sági et al. 2020).

The structure of Islamic stocks is inherently different from conventional stock markets. Portfolios under Islamic equity are based on real economic activity, which may lead to stocks being less risky (Raza et al. 2016; Varga and Tálos 2016), more stable (Erdogan et al. 2020; Kenourgios et al. 2016; Paltrinieri et al. 2019), safe from turmoil, and tied to the promotion of real assets. However, concerning the new global health crisis, a considerable lack of attention has been given to Islamic stock markets and their reaction and volatility.

From the health crisis to the economic turmoil, COVID-19 has impacted almost every sphere of life; the financial markets are no exception. So far, researchers have only focused on conventional stock markets and the Islamic stock markets have been given limited attention in connection with the current health crisis (see Sherif 2020). In the wake of the COVID-19 crisis, to the best of our knowledge, this study provided the first empirical evidence of volatility and its persistence in Islamic stock markets globally. The study aimed to examine the impact of the global health crisis on nine Islamic stock indices selected from around the world. Furthermore, it also intended to provide evidence of how long this shock is likely to persist in the Islamic stock markets.

The study was intended to add to the existing literature in two ways. First, it analyzed the effects of the COVID-19 crisis on Islamic stock markets around the globe, as there has been scant empirical evidence provided on this. Second, it contributed to the existing body of knowledge in terms of risk and volatility both before and after the outbreak. Thus far, the length of volatility shocks has only been analyzed on conventional stock markets. This study provided evidence on Islamic stock indices around the globe. Additionally, this study examined whether the global Islamic stock markets would be immune to the COVID-19 crisis, as they are intrinsically different from conventional stock markets.

The rest of the paper is organized as follows. The review of literature is organized and summarized in Section 2. Section 3 provides the data and methodology used in this study. as well as the econometric model specifications for this paper. Section 4 provides the results and discussion and explains how Islamic stock markets were affected by the COVID-19 crisis and the accompanying volatility shocks. Section 5 concludes the paper.

## 2. Literature Review

In the recent outbreak of COVID-19, to stop further transmission of virus, countries implemented lockdowns either at the national or local level. Consequently, the health crises caused global economic turmoil. The spread of COVID-19 affected stock markets in 21 different countries, which produced negative, abnormal returns soon after this outbreak (Al-Awadhi et al. 2020; Liu et al. 2020). The authors also stressed that the effect was worse in Asian stock markets as they could not implement strict health controls set up by their governments. Furthermore, Ashraf (2020) argued that stock markets reacted negatively as the number of cases rose in the first two months of the crisis. Additionally, stock market reactions from 25 different affected countries have been studied by Phan and Narayan (2020), who found that market overreactions were caused by news concerning COVID-19. However, the authors further argued that countries proactively responded to news of the pandemic, resulting in restrictions in travel, social gathering, and economic lockdowns across the globe. Although countries enforced several restrictions to control the spread of this virus, it could not be curbed completely.

In this short period of time, few studies have been conducted to capture the how the conventional stock market behaved in response to the coronavirus. For instance, Mazur et al. (2020) argued for an unpredictable reaction of different sectors in response to COVID-19, and found some showed resilience while others remained precarious. It was evident that the hospitality, petroleum, entertainment, and real estate sectors have been hit badly by the global pandemic. However, healthcare, service sector, and software industries enjoyed huge increases in market capitalization. A recent study by Zhang et al. (2020) documented evidence of a strong negative correlation between stock market returns and the number of infected people in ten countries with a high rate of infection. Furthermore, the study argued that stock markets become highly volatile and risky due to the pandemic and economic losses.

Topcu and Gulal (2020) using Driscoll–Kraay methods examined the impact of COVID-19 on Asian, European, and emerging stock markets. The results indicated that Asian stock markets were badly affected by the global health crisis. However, the study further argued that countries who took timely measures managed to weaken the adverse consequences of the corona virus outbreak. Additionally, the European stock markets were least affected compared to Asian and emerging markets. Similarly, Ashraf (2020) showed that with the increase of virus cases, the stock market became volatile and market returns went down proactively. The study considered 64 stock markets around the globe and found that stock market returns declined in line with the daily increase in the number of cases. Moreover, the author further argued that, in response to the pandemic, stock markets showed a quick response depending upon the intensity of the outbreak in that region. Additionally, Mirza et al. (2020) documented the evidence of a reduction in market capitalization of more than 12,000 non-financial companies in 15 European countries after the COVID-19 pandemic broke out. The authors argued that a global health emergency favored vulnerability in the corporate sector and the capital market. Similarly, Akrofi and Antwi (2020) studied the African energy sector and found a considerable downturn after the COVID-19 pandemic. Furthermore, Jeribi and Manzli (2021) analyzed the post-COVID-19 behavior of the Tunisian stock market and found a strong relationship between the number of confirmed cases (and deaths), and stock market performance. In essence, the authors found that as the number of confirmed cases and deaths increased the stock market performance in almost every sector became vulnerable. Vulnerability, negative shocks, and the dwindling trends of market indices have been documented by many authors so far. However, ever since Islamic stocks became part of financial markets, the research in this sector has been very limited regarding external shocks. Furthermore, it is of great importance to study the reaction of Islamic stock indices globally in response to the COVID-19 crisis.

Assessing the uncertainties relating to the investment portfolio is of paramount importance for financial assets. Uncertainty or risk associated with the financial time series is

referred as volatility, which is an essential component for the smooth functioning of the market, especially from an investor's perspective. The impact of COVID-19 is not only limited to a downturn of performance and economic activities; it also makes financial assets more prone to risk. Similarly, volatility is found to be more persistent in the post-COVID-19 period in conventional stock markets (Fakhfekh et al. 2021). The authors used GARCH family models and opined that investors must secure themselves from future volatility shocks through hedging tools. Likewise, using the EGARCH method, Insaidoo et al. (2021) found similar evidence in the Ghana stock index. The presence of a longer volatility persistence and asymmetric affect is evident in the Ghana stock index, which also highlighted the adverse impact of COVID-19 crisis on market volatility. The US stock market's volatility behaved in a similar fashion in response to the global health crisis (Baig et al. 2021).

Literature has been enriched with recent empirical evidence indicating a high proportion of volatility persistence irrespective of region. For instance, Guru and Das (2021) found a 69% increase in volatility in the Indian stock market. Albulescu (2021) investigated an increase in volatility in the Dow Jones and S&P500 and offered four key contributions to the literature. Firstly, the US stock market volatility increased with the increase in the number of cases globally. Secondly, the fatality rate appeared to have a positive impact on the volatility of the US stock market, which amplified risk in the market. Furthermore, the authors argued that the COVID-19 pandemic caused a greater detrimental effect on the US stock market compared to other crises in history. Nguyen et al. (2021) examined the volatility changes and volatility persistence of US and Chinese stock markets in pre- and during pandemic periods. The authors used the GARCH (1, 1) model and estimated that volatility persistence is similar for both stock markets. However, they further endorsed a weak spillover effect flowing from US to Chinese stock markets. Izzeldin et al. (2021) studied the stock markets in seven developed countries by sector and found that the UK, US, and Chinese stock markets experienced high volatility after the COVID-19 pandemic. Using GARCH (1, 1), the study suggested that the equity markets in China, the US, and UK suffered longer volatility persistence during the COVID-19 crisis. On the other hand, Engelhardt et al. (2021) conducted a multi-country analysis of 47 national stock markets based on societal trust in the government. The authors argued that in the countries with a high trust factor, market volatility (in reaction to the corona virus pandemic) is less than in low-trust countries. Furthermore, the dreadful impact of the pandemic has not only been observed in the corporate sector, stock market, and other financial assets but has also affected the volatility of bitcoin (Corbet et al. 2021). Concerning the Islamic stock index, Bahloul and Khemakhem (2021) studied the connectedness of returns and volatility of commodities and emerging MSCI Islamic stock indices using the VAR decomposition matrix. The degree of return and volatility spillover varied over time, but the authors claimed there was a strong transmission from commodities to MSCI Islamic indices in the post-COVID-19 period. However, the effect on volatility persistence due to the current pandemic on Islamic stock markets has not been explored.

## 3. Data and Methods

Thomson Reuter introduced the ideal ratings (IR) Islamic index in April 2009. These indices include the Islamic Australia index, Islamic BRIC index, Islamic MENA, MENASA, GCC, Bahrain, Malaysia, Kuwait, Oman, Qatar, Turkey, Emerging markets, and the ASEAN indices. According to Thomson Reuter's screening criteria, stocks must be in conformity with AAOIFI (Accounting and Auditing Organization for Islamic Financial Institutions) standards to be considered for the relevant index. Table 1 shows the list of Thomson Reuters Islamic indices that are included in this study.

**Table 1.** Selected Islamic Index.

| Indices | Abbreviation | Country/Region |
|---|---|---|
| Thomson Reuters IdealRatings Islamic Australia Index | ISAUSX | Australia |
| Thomson Reuters IdealRatings Islamic Qatar Index | ISQAX | Qatar |
| Thomson Reuters IdealRatings Islamic UAE Index | ISUX | United Arab Emirates |
| Thomson Reuters IdealRatings Islamic ASEAN Index | ISASEANX | ASEAN region |
| Thomson Reuters IdealRatings Islamic BRIC Index | ISBRICX | BRIC countries |
| Thomson Reuters IdealRatings Islamic GCC Index | ISGCCX | GCC |
| Thomson Reuters IdealRatings Islamic MENA Index | ISMENAX | MENA Countries |
| Thomson Reuters IdealRatings Islamic MENASA Index | ISMENSASX | MENASA Countries |
| Thomson Reuters IdealRatings Islamic Bahrain Index | ISBAHX | Bahrain |

Thomson Reuters' IdealRatings Islamic global index is considered as a market reflection. Furthermore, it is used as a benchmark to estimate the expected and abnormal returns of selected indices. Data are collected from the Thomson Reuters Eikon data stream. The daily closing values of selected Islamic stock indices were collected from 1 January 2019 to the end of 10 August 2020 from the mentioned source.

To address our research questions, the data are divided in two ways. Firstly, to study the market reaction, the data is grouped in three windows, i.e., estimation window, event window, and post-event or long-term window. However, to capture the volatility changes due to shocks, the data are divided into three periods: a full-length period, pre-crisis period, and post-crisis period.

### 3.1. Methodology

The analysis of stock market reactions in response to COVID-19 is conducted using the following two methods:

1. Event study methodology
2. Generalized autoregressive conditional heteroskedasticity (GARCH)

### 3.1.1. Event Study Methodology

The event study methodology is used to capture the effect of COVID-19 on the Islamic stock markets through selected indices. According to the efficient market hypothesis, if the market is efficient, the stock index will reflect the real change in value corresponding to a particular event or information disclosure (Fama et al. 1969; MacKinlay 1997). The market model is implemented using the selected data set, and further data are divided into three windows, i.e., estimation window (−1, −170), event window (0, 15), and post-event or long-term window (16, 43).

We selected 30th January 2020 as an event date as the WHO declared COVID-19 as a global health emergency on this day. To analyze the short window market reaction, we chose 16 days from 31 January to 15 February 2020 (0, 15). The second window was selected for the next 28 days, i.e., (16, 43), from 16th February to 31 March 2020. The last date of the second event window is the date when the WHO confirmed the spread of COVID-19 to 200 countries. Furthermore, for estimating the markets' expected returns, we selected an estimation window of 170 days prior to the event date, as studies suggest 100 to 250 days for setting up the estimation window (Carow and Kane 2002; Cox and Peterson 1994; MacKinlay 1997). However, to test the sensitivity of our results, we also employed different

lengths of estimation windows of $(-1, -130)$, $(-1, -150)$, and $(-1, -180)$ to calculate the abnormal return, following Liu et al. (2020). Moreover, similar results were obtained with different lengths of estimation window.

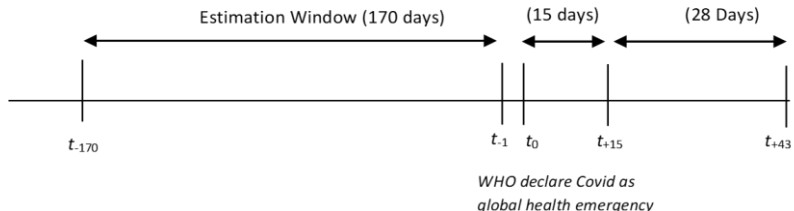

To estimate the significance of abnormal return on and after the event date, we used *t*-tests in this study. Furthermore, to ensure the robustness of the results, we also performed a nonparametric test, i.e., the Wilcoxon signed ranked test. Additionally, to analyze the market reaction in the short and long windows, we estimated cumulative abnormal returns (CARs) and cumulative averaged abnormal returns (CAARs).

We used Equation (1) to calculate the actual return. Furthermore, using the market model, we used OLS regression using Equation (2) to measure the market return.

$$R_{i,t} = \ln(I_{i,t}/I_{i,\,(t-1)}) \tag{1}$$

where $I_{i,t}$ represents the value of $i$ Islamic stock index at time $t$ and $R_{i,t}$ denotes the actual return.

$$\hat{R}_{i,t} = \delta_i + \gamma_i R_{mt} + \varepsilon_{i,t} \tag{2}$$

Expected return is denoted by $\hat{R}_{i,t}$. Each Islamic index intercept is represented as $\delta_i$ whereas $\gamma_i$ is the slope selected index at time $t$ with respect to market return, i.e., $R_{mt}$. The expected return calculated from Equation (2) using market model is differenced from the respective Islamic stock market return ($R_{i,t}$) to obtain the abnormal return of $i$ Islamic stock market. The difference between normal and expected return can be represented from Equation (3).

$$AR_{i,t} = R_{i,t} - \hat{R}_{i,t} \tag{3}$$

Equations (4)–(6) are used to calculate the average abnormal return (AAR), CARs, and CAARs, respectively.

$$AAR_i = \frac{1}{N} \sum_{i=1}^{N} AR_{i,\,t} \tag{4}$$

$$CAR_i(t_i) = \sum_{t=0}^{t(1\ to\ 43)} AR_{i,t} \tag{5}$$

$$CAAR_i(t_i) = \sum_{t=0}^{t(1\ to\ 43)} AAR_t \tag{6}$$

CAR and CAAR for short (0, 15) and long (16, 43) are calculated and *t*-test is performed to test the significance at 5%. Furthermore, all the statistics are given in the Tables 2–5 and Table A1.

$$t_{stat} = \frac{CAR_i}{Standard\ Error}$$

### 3.1.2. Generalized Autoregressive Conditional Heteroskedasticity (GARCH)

The conditional variance in the financial time series was first developed by Engle (1982). Autoregressive conditional heteroskedasticity shows the linearity between volatility changes and lagged squared residuals. The applicability of the ARCH model causes a limitation based on over-parameterization to capture consistent results, which is improved by Bollerslev (1986). The model was refined into a generalized ARCH, explaining the dependence

of conditional variance on its past squared residuals as well as past value of variances in the time series. Furthermore, the GARCH (1, 1) model successfully captures the tendency of volatility clustering and its persistence in the financial data (Choudhry 1996).

Market volatility of the chosen Islamic stock index is modelled through GARCH (1, 1) following the structure used in Choudhry (1996). The data extracted from Thomson Reuters is grouped into three categories, i.e., total sample, pre-COVID, and post-COVID. The total sample period starts from 1 January 2019 to 10 August 2020. The pre-COVID-19 period is taken from 1 January 2019 to 30 January 2020, and last period is chosen from 31 January 2020 to 10 August 2020. The day 30 January 2020 is when the global health emergency was declared by the WHO and is thus taken as the start of the COVID-19 crisis. Concerning the econometric model, a seminal work has been part of the empirical literature where the pre- and post-1987 crisis effect is modelled through GARCH (1, 1) with a limited sample size (Choudhry 1996). The study was conducted concerning 6 markets with 224 total observations (142 in the pre-crisis period and 82 in the post-crisis period). Furthermore, GARCH (1, 1) produced efficient results with the compliance of set restrictions (i.e., $\delta_i > 0$, $\alpha_i \geq 0$, and $\beta_i \geq 0$). A similar model setting has been used in other studies as well (see Choudhry et al. 2015; Rastogi 2014), to study the post-crisis period.

The conditional mean equation is obtained using ordinary least square regression, with the linear function of lagged return of itself. Furthermore, the stationarity of time series suggests that the market returns have constant means and constant variances over time. For instance, if the variance of the time series does not vary over time, a constant variance model is suggested to find the conditional variance in the time series. (Engle 1982). The stock market return is functioned by the lag of itself as represented in Equation (7). Whereas the variance Equation (9) represents GARCH (1, 1) model.

$$R_{i.t} = \mu_t + \theta_i R_{i.t-1} + \epsilon_t \tag{7}$$

$$\frac{\epsilon_t}{y_{t-1}} \sim N(0, \sigma_t) \tag{8}$$

$$\sigma_t = \delta_i + \sum_{i=1}^{x} \alpha_i (\epsilon_{t-i})^2 + \sum_{i=1}^{y} \beta_i \sigma_{t-i} \tag{9}$$

The stock market return is denoted by $R_{i.t}$. The mean of the return is represented as the intercept of the Equation (7), i.e., $\mu_t$. Equation (7) shows that the market return is conditional on previous day return, it further shows how well market value is predicted by its prior value, i.e., $R_{i.t-1}$. $\epsilon_t$ is the white noise of the system which is dependent on the past value and normally distributed with zero mean and time-varying variance, represented in Equation (8). GARCH (1, 1) models the variance of the system as a function of square lagged residuals and lagged to itself. Conditional variance is denoted as $\sigma_t$, Equation (9) shows that variance is conditional on lagged square of residuals and lagged of variance itself. The parameters of GARCH (1, 1) must comply with the given restrictions (i.e., $\delta_i > 0$, $\alpha_i \geq 0$, and $\beta_i \geq 0$), in order to validate the model and ensure positive volatility/variance. In contrast to ARCH (p), GARCH (1, 1) is more parsimonious as it only has three parameters, which are $\delta_i$, $\alpha_i$, and $\beta_i$. In particular, the magnitude and the significance of $\alpha_i$ represent the ARCH effect of the selected stock return. It measures how stock return volatility changes with the lagged shocks (i.e., lagged squared residuals). On the other hand, $\beta_i$ represents the volatility changes by the momentum within the system itself, i.e., the lagged variance of the index return. The $\beta_i$ captures the volatility clustering within market returns, which refers to how a period of high volatility tends to follow a period of low volatility (Bollerslev 1986).

However, GARCH (1, 1) mode is parsimonious and efficient in estimating volatility clustering of financial time series without violating the restrictions (i.e., $\delta_i > 0$, $\alpha_i \geq 0$, and $\beta_i \geq 0$) (Tsay 2002). Furthermore, GARCH (1, 1) is expected to produce efficient estimates even with a small sample size from 150 to 200, provided that the positivity constraints of the parameter are not violated (David and Ampah 2018; Leong 2018; Lumsdaine 1995).

Furthermore, as the number of observations falls below 150, the parameters of GARCH (1, 1) violate the positivity restriction. Additionally, the boundary proportion becomes close to unity, which is seen as $\alpha_i$ becomes close to zero (Leong 2018). Estimating the volatility persistence of a stock index can be measured through the sum of ARCH and GARCH coefficients. If $\alpha_i + \beta_i = 1$, it implies that a shock in the financial time series is going to persist for an indefinite period causing volatility persistent for a longer future period (Engle and Bollerslev 1986). However, if the sum of ARCH and GARCH coefficients approaches unity, the volatility persistence is greater (Poterba and Summers 1986). In other words, the sum of $\alpha_i + \beta_i$ represents the response function of the shocks to volatility per period. A value significantly less than unity would show a weaker persistence of volatility, which will decline in the near future, whereas a value close to 1 indicates that the market shock will last for a longer period. However, a value more than 1 will indicate that shocks will have indefinite persistence volatility for an undefined period (Chou 1988; Choudhry 1996).

## 4. Results and Discussion

### 4.1. Islamic Stock Market Reaction

Table 2 shows the descriptive statistics of the data in two groups (pre- and post-event groups). According to the mean and standard deviation of the pre-event period (170 trading days before the date of event) of all selected indices, it is evident that the mean is positive, except in the Islamic Bahrain index (ISBAHX), i.e., $-0.0001$ with st. dev. 0.0114. However, the returns of all indices except that of ISBAHX, do not deviate much from the mean. The minimum dispersion is observed in the Islamic stock GCC index with a standard deviation of 0.0048 and mean value of 0.0003. The maximum mean deviation in the pre-event group is observed in the Islamic stock Australia index (i.e., 0.0078). The mean return from the event date till 31 March 2020 (43rd day post-event) appeared as a negative mean with considerably higher mean dispersion compared with pre-event statistics. The negative mean implies that a negative return occurred in all the selected Islamic stock markets globally. The least percentage decrease happened in the MENA region with a 514% decrease in the mean value from pre- to post-event periods.

**Table 2.** Differences in mean returns of sample indices.

| Index | Number of Trading Days | Event Group's Mean | Event Group's Std. Dev. |
|---|---|---|---|
| **Pre-Event Mean and St Deviation** | | | |
| ISAUSX | 170 | 0.0007 | 0.0078 |
| ISQAX | 170 | 0.0001 | 0.0058 |
| ISUX | 170 | 0.0006 | 0.0066 |
| ISASEANX | 170 | 0.0001 | 0.0057 |
| ISBRICX | 170 | 0.0005 | 0.0072 |
| ISGCCX | 170 | 0.0002 | 0.0048 |
| ISMENAX | 170 | 0.0013 | 0.0068 |
| ISMENSASX | 170 | 0.0006 | 0.0071 |
| ISBAHX | 170 | $-0.0001$ | 0.0114 |
| **Post-Event Mean and St Deviation** | | | |
| ISAUSX | 43 | $-0.0067$ | 0.0338 |
| ISQAX | 43 | $-0.0057$ | 0.0179 |
| ISUX | 43 | $-0.0073$ | 0.0280 |
| ISASEANX | 43 | $-0.0073$ | 0.0283 |
| ISBRICX | 43 | $-0.0078$ | 0.0326 |
| ISGCCX | 43 | $-0.0062$ | 0.0204 |
| ISMENAX | 43 | $-0.0065$ | 0.0246 |
| ISMENSASX | 43 | $-0.0066$ | 0.0286 |
| ISBAHX | 43 | $-0.0030$ | 0.0116 |

Table 3 represent the abnormal returns of all the indices on the event day, 1 day, and 2 days after the event day, represented as $t_0$, $t_{+1}$, $t_{+2}$, respectively. On 30 January 2020, as news broke out from China, the Islamic stock indices of Australia, UAE, ASEAN region, BRIC, GCC, MENA, MENASA, Bahrain reflected a negative return on the first day. Qatar was the only country which remained stable as an Islamic stock market. However, on the following day, the market reacted in a mixed manner with growth in the Islamic index of Australia at 0.4%, UAE at 0.12%, and MENA at 0.1%. In contrast, there was –1.2% return on the ASEAN Islamic stock market, i.e., a negative return. On the second day after the WHO announced the COVID-19 global emergency, the Islamic stock markets dropped by 1.3%, 1.0%, 0.8%, 0.77%, 1.8%, 0.4%, and 0.1% in Australia, Qatar, UAE, ASEAN region, GCC, MENA, MENASA, and Bahrain, respectively.

**Table 3.** Abnormal return on event date and one day after.

| Index | Abnormal Return | | |
|---|---|---|---|
| | Event Day $(t_0)$ | 1 Day after Event Day $(t_{+1})$ | 2 Day after Event Day $(t_{+2})$ |
| ISAUSX | −0.0101 | 0.0041 | −0.0131 |
| ISQAX | 0.0008 | −0.0075 | −0.0100 |
| ISUX | −0.0017 | 0.0013 | −0.0080 |
| ISASEANX | −0.0076 | −0.0119 | −0.0139 |
| ISBRICX | −0.0154 | −0.0034 | 0.0019 |
| ISGCCX | −0.0009 | −0.0046 | −0.0078 |
| ISMENAX | −0.0019 | 0.0011 | −0.0182 |
| ISMENSASX | −0.0081 | −0.0006 | −0.0044 |
| ISBAHX | −0.0048 | 0.0038 | −0.0010 |

Figure 1 represents the graphical plot of abnormal returns (AR) of selected indices. The graph shows the daily abnormal return from 11 days to 43 days until the event. Furthermore, the returns proved to be more volatile and diffuse after the first short event window. The reason for more dispersed AR is the spread of virus in almost all regions of the world including GCC, MENA, ASEAN, and other countries as well. Australia was one of the countries where the first case appeared on 25 January 2020. According to the WHO situation report–71, by the start of March, COVID-19 cases started to grow exponentially in more than 200 countries. Consequently, its effect started to emerge in terms of negative and fluctuating abnormal returns in all Islamic stock markets around the globe. A more accurate picture of COVID-19 crisis is shown in Figure 2. CAR from −11 days to 43 days represents how quickly markets declined around the globe. From day 0 to 31 March, the Islamic stock indexes of Australia, MENA, BRIC, MENASA, UAE, and ASEAN dropped by 27.32%, 31.53%, 23.7%, 26.16%, 29.18%, and 26.16%, respectively. However, the market return for the Bahrain index proved to be most stable among the rest of the Islamic indices. The reason behind this stability appears to be a smaller number of cases till 31 March 2020. In Bahrain, there were only 569 total cases of COVID-19 by the end of March. Furthermore, the Bahrain stock index was the least affected by the COVID-19 pandemic.

Figure 2 represents the cumulative abnormal returns (CAR) of all selected Thomson Reuters Islamic indices. We could draw the conclusion that in most countries or regions the outbreak of the pandemic had occurred after mid-February or the beginning of March 2020. However, the markets responded accordingly soon after the first confirmed COVID-19 case appeared. Furthermore, Islamic stock indices from MENA, MENASA, and BRIC produced negative returns after the short-term window. The outbreak in the region of MENA, MENASA, and BRIC happened as early as 31 December 2019 in China and as late as 1 April in Yemen. As a result, news related to COVID-19 directly affected the investor's sentiments and hence the market reaction.

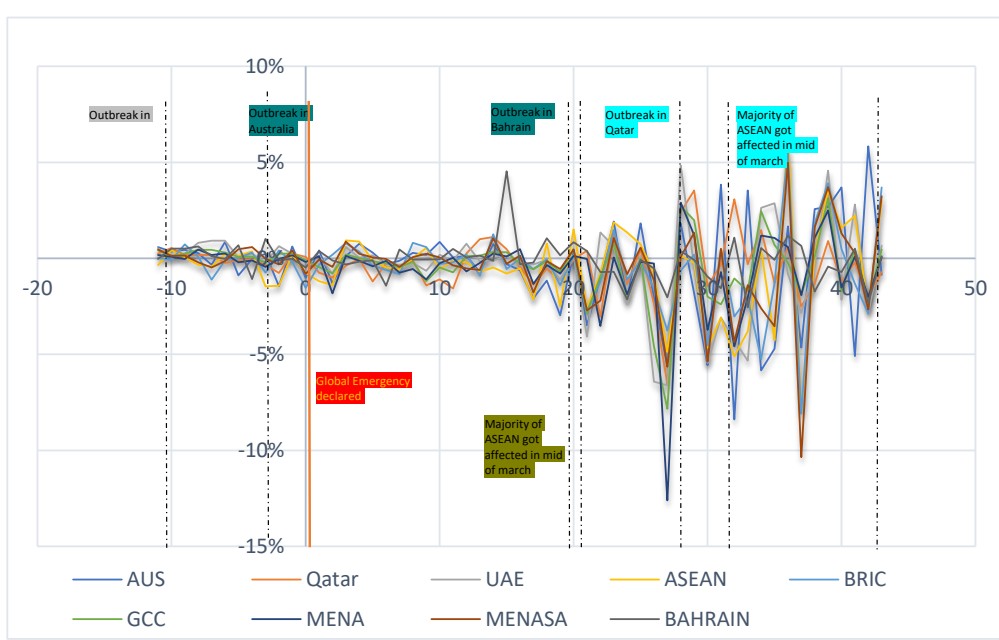

**Figure 1.** Abnormal return (AR) change of Thomson Reuters Islamic stock indices.

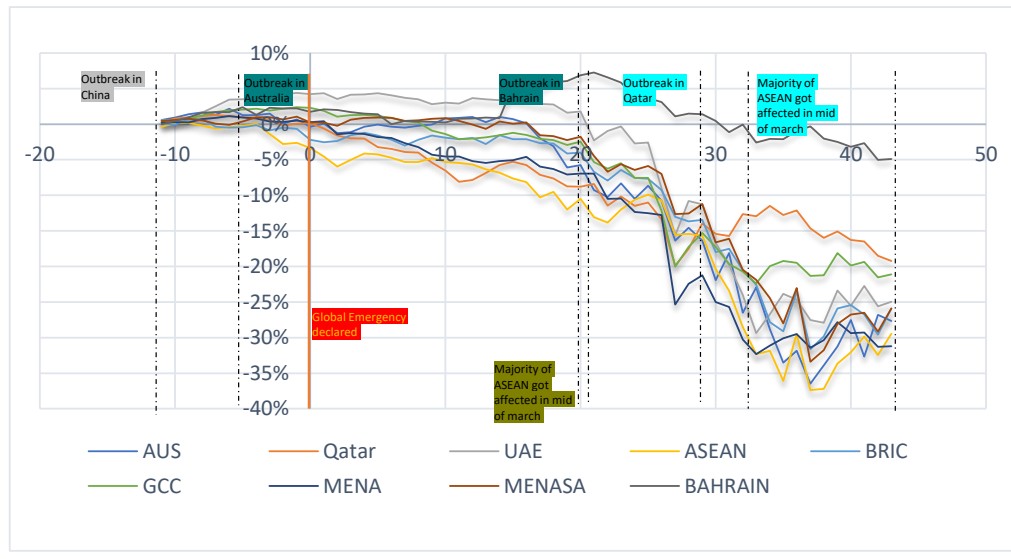

**Figure 2.** Cumulative abnormal return of Islamic stock markets after event date.

Tables 4 and 5 show the cumulative abnormal returns (CAR) and their level of significance in the short- and long-term event window, respectively. The results show that the CAR of the Islamic Australia index is positive and insignificant, whereas the CAR of the Islamic GCC index is insignificant and negative, which indicates that in the short run ISGCCX and ISAUSX were not significantly affected by COVID-19. However, the value of the CAR is negative but not statistically significant. On the contrary, the CAR of the Islamic Qatar index is −0.0415 significant at 1%, which shows sudden downturn effects soon after the pandemic broke out. In contrast, the CAR of UAE, MENASA, and Bahrain were significantly positive in the short event window (0, 15) at 5% level of significance. During this time (0, 15), the Arab and MENASA (Middle East, North Africa, South Asia) regions were not overcome by COVID-19 cases.

**Table 4.** Cumulative abnormal return (CAR) in short-term window.

| Index | CAR i (0–15) | *t*-Test | *p* Value | Willcoxon-Stat | *p* Value |
|---|---|---|---|---|---|
| ISAUSX | 0.0003 | 0.1737 | 0.3856 | -0.052 | 0.9588 |
| ISQAX | −0.0415 * | −6.3692 | 0.0000 | −3.464 | 0.0005 |
| ISUX | 0.0368 * | 28.2823 | 0.0000 | 3.516 | 0.0004 |
| ISASEANX | −0.0529 * | −19.3117 | 0.0000 | −3.516 | 0.0004 |
| ISBRICX | −0.0206 * | −16.4032 | 0.0000 | −3.516 | 0.0004 |
| ISGCCX | −0.0004 | −0.0935 | 0.3901 | −0.310 | 0.7564 |
| ISMENAX | −0.0295 * | −5.7347 | 0.0001 | −3.361 | 0.0008 |
| ISMENSASX | 0.0039 * | 3.3419 | 0.0048 | 2.689 | 0.0072 |
| ISBAHX | 0.0127 * | 3.8978 | 0.0016 | 3.464 | 0.0005 |

Note: * significant at 5%.

**Table 5.** Cumulative abnormal return in long-term window.

| Index | CAR i (16–43) | *t* Test | *p* Value | Willcoxon-Stat | *p* Value |
|---|---|---|---|---|---|
| ISAUSX | −0.1864 * | −8.7369 | 0.0000 | −4.600 | 0.0000 |
| ISQAX | −0.1302 * | −18.3111 | 0.0000 | −4.623 | 0.0000 |
| ISUX | −0.1379 * | −6.1274 | 0.0000 | −3.848 | 0.0001 |
| ISASEANX | −0.2147 * | −10.9212 | 0.0000 | −4.623 | 0.0000 |
| ISBRICX | −0.1627 * | −8.5449 | 0.0000 | −4.602 | 0.0000 |
| ISGCCX | −0.1390 * | −9.6128 | 0.0000 | −4.620 | 0.0000 |
| ISMENAX | −0.2071 * | −10.6571 | 0.0000 | −4.623 | 0.0000 |
| ISMENSASX | −0.1542 * | −7.4663 | 0.0000 | −4.600 | 0.0000 |
| ISBAHX | 0.0121 | 1.6740 | 0.0992 | 1.457 | 0.1450 |

Note: * significant at 5%.

The Islamic stock portfolios of ASEAN, BRIC, GCC, MENA showed significantly negative CAR of −0.0529 (at 5% level), –0.0206 (at 5% level), –0.0004 (at 5% level), and −0.0295 (at 5% level), respectively. The results suggest that five out of nine selected Islamic stock indices reacted immediately to the news of the pandemic in the short event window (0, 15). When the COVID-19 spread out to almost every country, the Islamic stock markets reacted in the same way as conventional markets. Our findings are consistent with Sherif (2020), who asserted that the Islamic stock market reacted the same way as the conventional market. However, the Islamic Bahrain index was least affected by the crises (CAR of 0.012 at 5% significance). As shown in Figures 1 and 2, a decline is observed after the first case appeared in Bahrain on 21 February 2020. Even then, the respective Islamic index remained stable compared to other indices. CAR of the Islamic index of Australia is recorded as −0.1864 (5% level), Qatar at −0.1302 (5% level), UAE at -0.1379 (5% level), ASEAN at −0.2147 (5% level), BRIC at −0.1627 (5% level), GCC at −0.1390 (5% level), MENA at −0.2072 (5% level), and MENASA at −0.1542 (5% level) in the long-term event window (16, 43). By this time, the whole world had encountered COVID-19. The results of this paper are consistent with the findings of Sherif (2020).

To see the overall market reaction on the post-event date, cumulative averaged abnormal returns (CAARs) were calculated across all nine selected indices. Table A1 represents the value of daily CAAR and its significance is given on Table 5 from day 0 to day 43. On the event day, the CAAR is positive and significant at 5%; it remained positive but insignificant on day 1 as well. It is evident that the Islamic stock market remained stable on the day the WHO declared COVID-19 as a health emergency. However, a negative CAAR of –0.68% significant at 5% is observed on day 2 of the event, which increased to its highest value of −26% (significant at 5%) on day 37 and ended at −23% (significant at 5%) on the 43rd day of the event. However, previous findings concerning the immunity of Islamic Stocks from external crises (Erdogan et al. 2020; Kenourgios et al. 2016; Paltrinieri et al. 2019) is not supported by our results. Unlike other crises, this pandemic forcibly required countries to halt their economic activities, set up lock downs, close borders, and limit social or business meetings. Additionally, the essence of the Islamic stock index is based on real economic output. Thus, if the economic activity is barred due to lockdown, then producing a positive return is almost impossible.

Averaged abnormal returns (AARs) and cumulative averaged abnormal returns (CAARs) of nine selected indices are displayed in Figure A1. Until the 15th day of the event, the market's AAR remained stable. Although there is a gradual downturn in the outcome of CAAR since the disease broke out, a significant decline is evident from mid-February 2020. Consequently, the volatility of AAR is much higher in the long-term window (16, 43); hence, there is a drastic decline in CAAR from 0.68% to −23.3%.

### 4.2. Islamic Stock Market Volatility
#### 4.2.1. Descriptive Statistics

Table 6 illustrates the descriptive statistics of all three periods of the data under study. The mean of the Islamic ASEAN, Australia, and UAE indexes are negative in the full periods, which shows that investors suffered losses more than gains in that period. The standard deviation figures represent the fluctuations in daily returns of all selected Islamic indices. Furthermore, a high value of kurtosis is found in all market indexes, which is more than the normal value (3). The time series of selected stock indexes is not normally distributed, as is evident from Jarque–Bera statistics, which is significant at 1% in all three periods. Additionally, the data series is fat tailed (leptokurtic), as represented by the higher-than-normal value of kurtosis, and the time series of all periods are found to be static at that level. However, the data satisfies the preconditions of ARCH/GRACH modelling for estimating volatility and its persistence (Choudhry 1996; Engle and Bollerslev 1986; Li et al. 2002).

**Table 6.** Descriptive statistics.

| | Mean | Median | SD | Kurtosis | Jarqu Bera | ADF |
|---|---|---|---|---|---|---|
| **Full Period (1 January 2019–10 August 2020)** | | | | | | |
| ASEAN | −0.00001 | 0.00086 | 0.01153 | 18.8732 | 5113.65 * | 0.0000 |
| AUS | 0.00045 | 0.00169 | 0.01422 | 13.3198 | 2204.84 * | 0.0000 |
| BAHRAIN | −0.00001 | 0.00000 | 0.01707 | 8.01910 | 499.973 * | 0.0000 |
| BRIC | 0.00023 | 0.00098 | 0.01320 | 19.7328 | 5694.52 * | 0.0000 |
| GCC | 0.00021 | 0.00016 | 0.00866 | 34.3845 | 19,772.4 * | 0.0000 |
| MENA | 0.00054 | 0.00051 | 0.00996 | 72.7178 | 95,591.6 * | 0.0000 |
| MENASA | 0.00009 | 0.00095 | 0.01417 | 48.5166 | 41,394.9 * | 0.0000 |
| QATAR | 0.00326 | 0.00023 | 0.06557 | 442.894 | 375,837 * | 0.0000 |
| UAE | −0.00002 | 0.00000 | 0.01330 | 18.6795 | 5087.77 * | 0.0000 |
| **Pre-Crisis Period: (1 January 2019–30 January 2020)** | | | | | | |
| ASEAN | 0.00016 | 0.00079 | 0.00547 | 4.03954 | 21.0587 * | 0.0000 |
| AUS | 0.00081 | 0.00167 | 0.00724 | 5.29287 | 102.662 * | 0.0000 |
| BAHRAIN | −0.00079 | 0.00000 | 0.01378 | 6.18449 | 143.346 * | 0.0000 |
| BRIC | 0.00036 | 0.00082 | 0.00699 | 5.41240 | 69.8254 * | 0.0000 |
| GCC | 0.00029 | 0.00000 | 0.00520 | 10.2327 | 688.815 * | 0.0000 |
| MENA | 0.00084 | 0.00039 | 0.00620 | 19.8784 | 3573.84 * | 0.0000 |
| MENASA | 0.00036 | 0.00055 | 0.00681 | 9.15511 | 457.737 * | 0.0000 |
| QATAR | 0.00023 | 0.00000 | 0.00664 | 4.96006 | 49.9837 * | 0.0000 |
| UAE | 0.00036 | 0.00000 | 0.00780 | 10.5451 | 704.779 * | 0.0000 |
| **Post-Crisis Period: (31 January 2020–10 August 2020)** | | | | | | |
| ASEAN | −0.00027 | 0.00147 | 0.01718 | 9.73290 | 397.005 * | 0.0000 |
| AUS | −0.00011 | 0.00178 | 0.02092 | 7.10878 | 157.552 * | 0.0000 |
| BAHRAIN | 0.00121 | 0.00000 | 0.02122 | 6.70326 | 122.217 * | 0.0000 |
| BRIC | 0.00002 | 0.00175 | 0.01928 | 10.9754 | 554.307 * | 0.0032 |
| GCC | 0.000078 | 0.00091 | 0.01227 | 21.3834 | 2763.91 * | 0.0039 |
| MENA | 0.00006 | 0.00074 | 0.01395 | 46.8239 | 15,163.5 * | 0.0084 |
| MENASA | −0.00034 | 0.00166 | 0.02107 | 25.2008 | 4053.97 * | 0.0000 |
| QATAR | 0.00802 | 0.00148 | 0.10471 | 173.534 | 223,220 * | 0.0000 |
| UAE | −0.00062 | 0.00020 | 0.01897 | 10.9904 | 569.489 * | 0.0000 |

* denotes significance at 1%.

In contrast, the mean returns of pre-crisis periods are positive, except for the Islamic Bahrain stock index. Additionally, the returns of the Bahrain stock index greatly deviate from the mean value. The deviation of returns from the mean return is more in the full period compared to the pre-crisis period. However, the time series of the pre-crisis period is leptokurtic and significantly deviated from normal distribution, as evident from Jarque Bera statistics. While considering the post-crisis period, the mean returns of ASEAN, Australia, MENASA, and UAE are negative with a high degree of deviation from mean returns, compared to the pre-crisis period. It shows that the stock market returns were highly volatile after the outbreak of the virus. Furthermore, the time series in post-crisis periods shows fat-tailed distributions and significant Jarque–Bera figures. The descriptive statistics for all periods show evidence of fat-tailed, leptokurtic, highly volatile, and abnormally distributed time series data. Therefore, the results support the application of the ARCH/GARCH model to estimate Islamic stock index volatility before and after the outbreak of COVID-19. Furthermore, Figure 3 shows the stock index trend for the full period of all nine Islamic stock indexes. Unlike Qatar, all stock indexes showed a sharp decline in the first two months of 2020. Furthermore, Figure A2 (Appendix B) shows evidence of volatility clustering in all indexes except the Qatar Islamic index. The period of high volatility and the period of low volatility tend to cluster in all stock indexes except Qatar. Figures A3 and A4 (Appendix B) show the stock index return illustration for pre- and post-crisis. Furthermore, high and low returns fluctuate and tend to cluster more clearly in the post-crisis period, except for the Qatar Islamic index. However, a constant and mean reverting variance can be observed in the GCC, MENA, and MENASA stock indexes for the pre-crisis period.

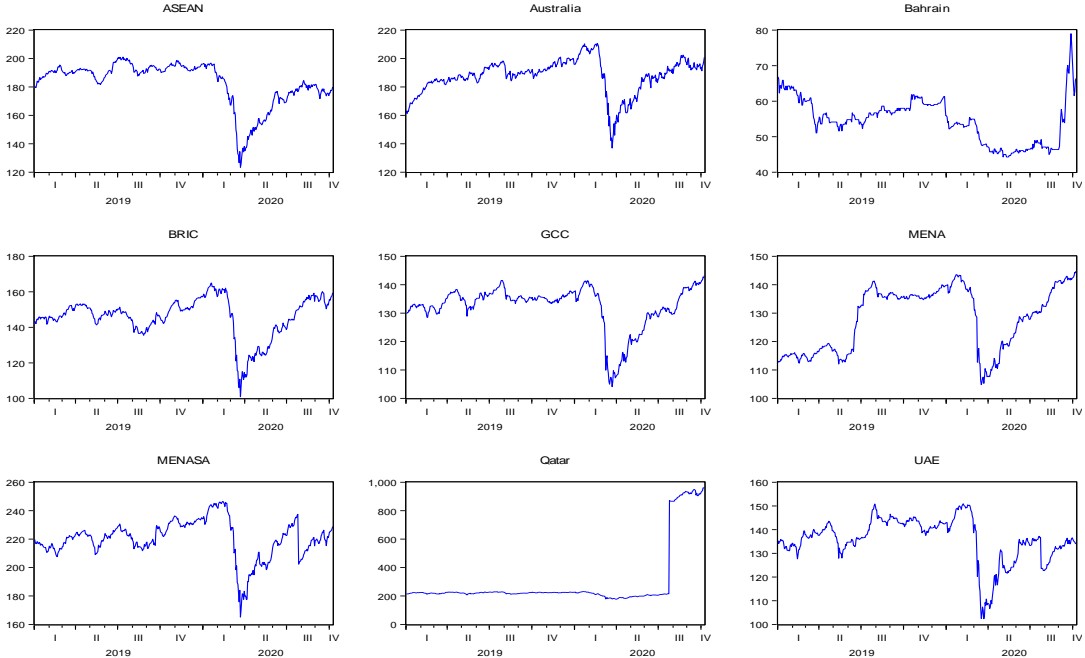

**Figure 3.** Market trend. Source: authors' calculation using EViews 10.

The presence of volatility clustering in full, pre-, and post-COVID-19 periods is evident from Figures A2–A4 (Appendix B). Additionally, Table 7 shows the test of stationarity of all returns in the full, pre-, and post-crisis periods. The augmented Dickey–Fuller (ADF) test statistics are significant at 1% for all Islamic stock indexes in all periods under study.

**Table 7.** GARCH (1, 1) full period.

| | $\mu_t$ | $\theta_i$ | $\delta_i$ $\times\, 10^{-5}$ | $\alpha_i$ | $\beta_i$ | $\alpha_i + \beta_i$ | LL [1] | ARCH-LM |
|---|---|---|---|---|---|---|---|---|
| ISASEANX | 0.0003 (0.839) | −0.0048 (−0.089) | 0.202 (2.511) [b] | 0.1611 (4.910) [a] | 0.8193 (22.82) [a] | 0.9804 | 1588.3 | 0.7307 |
| ISAUSX | 0.0010 (2.284) [b] | −0.0927 (−1.91) [c] | 0.452 (3.120) [a] | 0.1705 (5.313) [a] | 0.8008 (21.92) [a] | 0.9713 | 1472.1 | 0.6326 |
| ISBAHX | −0.0004 (−0.740) | −0.0186 (−0.259) | 5.37 (8.814) [a] | 0.3277 (7.737) [a] | 0.4880 (10.36) [a] | 0.8156 | 1304.7 | 0.6966 |
| ISBRICX | 0.0008 (2.040) [b] | 0.1396 (2.25) [b] | 0.424 (3.451) [a] | 0.1532 (6.444) [a] | 0.8123 (28.57) [a] | 0.9655 | 1512.7 | 0.9509 |
| ISGCCX | 0.0006 (1.883) [c] | 0.0419 (0.694) | 0.407 (5.324) [a] | 0.1864 (7.104) [a] | 0.7428 (19.69) [a] | 0.929 | 1683.6 | 0.7944 |
| ISMENAX | 0.0011 (0.826) | −0.0146 (−0.078) | 6.420 (1.892) [c] | 0.150 (0.980) | 0.599 (2.964) [a] | 0.749 | 1456.7 | 0.9470 |
| ISMENSASX | 0.0009 (3.26) [a] | 0.0579 (1.481) | 1.640 (1.921) [c] | 0.1690 (1.797) | 0.7618 (11.01) [a] | 0.931 | 1401.5 | 0.8949 |
| ISQAX | 0.0044 (0.1254) | −0.0050 (−0.005) | 0.0028 (0.837) | −0.0035 (−0.700) | 0.5937 (1.217) | 0.59722 | 586.70 | 0.9982 |
| ISUX | 0.0003 (0.513) | 0.0649 (0.908) | 2.370 (4.795) [a] | 0.1636 (3.538) [a] | 0.6816 (10.38) [a] | 0.8451 | 1426.0 | 0.9253 |

[a,b,c] denotes significance at 1%, 5%, and 10%; [1] LL: log likelihood value.

### 4.2.2. GARCH Results

This section provides the results of the GARCH (1, 1) model, which is explained in Section 3.1.2. The model estimates the mean equation through its lagged value and volatility changes in response to lagged shocks and momentum within the system. Tables 7–9 illustrate the results of the GARCH (1, 1) model for the whole period (2 January 2019 to 10 August 2020), pre-crisis period (2 January 2019 to 30 January 2020), and post-crisis period (31 January 2020 to 10 August 2020), respectively. The significance of the results is calculated on z-stats, which are given by the GARCH (1, 1) estimates using EViews 10. The presence of an ARCH effect in the full period is evident in all Islamic stock indices except the MENA, MENASA, and Qatar Islamic stock indexes. However, descriptive statistical evidence showed the presence of autocorrelation, fat-tailed, volatility clusters in all indices. However, a significant ARCH effect is present for the full period in ASEAN, Australia, BRIC, GCC, UAE, and Bahrain indexes.

The results show that the magnitude of $\alpha_i$ is less than unity, which implies that the results are not explosive. On the other hand, the presence of volatility clustering is significantly evident from the GARCH coefficient of all stock indices, except for the Qatar Islamic stock index. The GARCH (1, 1) model is selected based on the lowest AIC (Akaike Info criteria), maximum log likelihood and a higher value of adjusted R squared. However, we considered other model selection alternatives as well, which includes SIC (Schwarz info criteria) and HC (Hannan–Quinn criteria). AIC appeared to be the lowest figure among other choices. Furthermore, based on the nature of data used in this study, a univariate symmetric GARCH model has been applied in this study. Based on the model selection criteria, in the longest period, all selected time series were modelled with GARCH (1, 1) using normal gaussian error distribution. The value of $\beta_i$ for all stock returns is not more than unity, which satisfies the GARCH (1, 1) conditions to not be explosive. Furthermore, the ARCH-LM test is performed to diagnose the element of heteroscedasticity in the optimal selected model. Additionally, autocorrelation is estimated using a correlogram of squared residual using 24 lags giving no evidence of autocorrelation.

**Table 8.** GARCH (1, 1) pre-crisis period.

| | $\mu_t$ | $\theta_i$ | $\delta_i$ $\times 10^{-5}$ | $\beta_i$ | $\beta_i$ | $\alpha_i + \beta_i$ | LL [1] | ARCH-LM |
|---|---|---|---|---|---|---|---|---|
| ISASEANX | 0.0004 (1.362) | −0.0199 (−0.325) | 0.925 (1.070) [b] | 0.0927 (1.303) | 0.6075 (1.960) [b] | 0.699 | 1073.4 | 0.581 |
| ISAUSX | 0.0013 (3.09) [a] | 0.00197 (0.0310) | 0.941 (0.935) | 0.0451 (1.019) | 0.7672 (3.500) [a] | 0.812 | 999.2 | 0.133 |
| ISBAHX | 0.0002 (0.0059) | −0.0203 (−0.417) | 0.410 (2.675) [a] | 0.1608 (4.785) [a] | 0.7284 (18.62) [a] | 0.889 | 892.8 | 0.932 |
| ISBRICX | 0.0006 (1.524) | 0.1421 (2.24) [b] | 1.570 (0.950) | 0.1024 (1.218) | 0.5731 (1.466) | 0.675 | 1010.8 | 0.954 |
| ISGCCX | 0.0004 (1.066) | −0.0226 (−0.241) | 0.620 (1.567) | 0.0905 (1.938) [c] | 0.6854 (3.66) [a] | 0.776 | 1084.4 | 0.556 |
| ISMENAX | 0.0004 (6.81) [a] | −0.0184 (−1.102) | 0.329 (1.319) | 0.0657 (1.158) | 0.8302 (7.49) [a] | 0.895 | 1133.4 | 0.884 |
| ISMENSASX | 0.0005 (1.396) | 0.0659 (1.251) | 0.745 (0.840) | 0.0516 (0.852) | 0.786 (3.43) [a] | 0.838 | 1032.1 | 0.969 |
| ISQAX | 0.0003 (1.160) | 0.0400 (0.681) | 3.840 (8.902) [a] | 0.3166 (3.947) [a] | 0.205 (2.86) [a] | 0.522 | 1026.5 | 0.738 |
| ISUX | 0.0005 (0.866) | 0.0078 (0.095) | 1.270 (1.854) [c] | 0.0950 (2.028) [b] | 0.694 (4.56) [a] | 0.789 | 976.1 | 0.898 |

[a,b,c] denotes significance at 1%, 5%, and 10%; [1] LL: log likelihood value.

Comparisons of the volatility changes by shocks and the presence of volatility clustering before and after the pandemic are presented in Tables 8 and 9. The coefficients of ARCH in the pre-crisis period are only significant for Bahrain, Qatar, and the UAE. ASEAN, BRIC, GCC, MENA, MENASA, and Australian Islamic stock markets appear to be less volatile before COVID-19. In the post-crisis period, the ARCH effect is evident in all Islamic stock indices, except the MENA and Qatar Islamic stock indexes. However, a significant increase in the ARCH coefficient is observed from pre- to post-crisis periods, irrespective of the level of significance. The opposite is true from the MENASA and Qatar Islamic stock indexes. On the other hand, $\beta_i$ of selected stock indices is significant at 1 percent in the post-crisis periods. This implies that due to COVID-19 the market experienced high volatility. Furthermore, the volatility of the current period is significantly dependent on the volatility of the previous period. For instance, the ARCH effect significantly changed from 0.093 to 0.211 for the ASEAN Islamic stock index, from 0.045 to 0.189 for the Australian Islamic index, 0.161 to 0.71 for the Bahrain stock index, 0.10 to 0.183 for the BRIC region, 0.091 to 0.22 in GCC, and 0.065 to 0.077 for the MENA Islamic index. It is also worth noting that the ARCH effects after the COVID-19 break out in the Islamic stock indices were significant at 5%, except for the MENA, MENASA, and Qatar indexes. The GARCH coefficient of the post-crisis period has significantly increased in all stock returns except UAE, MENA, and Bahrain. However, the magnitude of $\beta_i$ has increased significantly from the pre- to post-crisis period. This implies that due to the pandemic the Islamic stock indexes tend to become more volatile and are dependent on its lagged volatility. Consequently, the GARCH effect is more evident in the post-COVID-19 period, where a period of high stock return follows the pattern of low stock returns in sequence. Furthermore, the optimal GARCH (1, 1) model is selected based on AIC, log likelihood, and adjusted R squared value. Remaining ARCH effects and autocorrelation are diagnosed by ARCH-LM and the correlogram R squared matrix.

**Table 9.** GARCH (1, 1) post-crisis period.

| | $\mu_t$ | $\theta_i$ | $\delta_i$ $\times 10^{-5}$ | $\alpha_i$ | $\beta_i$ | $\alpha_i + \beta_i$ | LL [1] | ARCH-LM |
|---|---|---|---|---|---|---|---|---|
| ISASEANX | 0.0010 (1.241) | −0.0048 (−0.052) | 0.984 (1.333) | 0.211 (2.253) [b] | 0.7473 (7.967) [a] | 0.958 | 529.15 | 0.3465 |
| ISAUSX | 0.0016 (1.493) | −0.231 (−2.78) [a] | 1.520 (1.394) [c] | 0.1893 (2.179) [b] | 0.7719 (7.845) [a] | 0.961 | 489.67 | 0.8383 |
| ISBAHX | 0.0004 (0.388) | 0.1823 (1.462) | 9.540 (6.901) [a] | 0.7091 (6.216) [a] | 0.2282 (2.975) [a] | 0.937 | 480.81 | 0.8572 |
| ISBRICX | 0.0020 (2.74) [a] | 0.0842 (1.005) | 0.887 (1.366) | 0.1835 (2.134) [b] | 0.7969 (10.94) [a] | 0.979 | 527.26 | 0.7814 |
| ISGCCX | 0.0016 (3.595) [b] | 0.0879 (0.876) | 0.299 (1.751) [c] | 0.2154 (3.237) [a] | 0.7427 (10.59) [a] | 0.957 | 595.32 | 0.4495 |
| ISMENAX | 0.0008 (3.38) [a] | 0.0087 (0.270) | 0.234 (1.171) | 0.0774 (1.475) | 0.8899 (15.78) [a] | 0.966 | 626.25 | 0.8872 |
| ISMENSASX | 0.0004 (0.016) | −0.099 (−0.674) | 6.880 (1.900) [c] | 0.0585 (1.802) | 0.790 (7.652) [a] | 0.848 | 444.76 | 0.8334 |
| ISQAX | 0.0031 (21.38) | −0.005 (−0.244) | 0.0002 (0.227) | −0.0002 (−1.031) | 0.5195 (0.244) | 0.519 | 539.21 | 0.9386 |
| ISUX | 0.0002 (0.101) | 0.1302 (1.164) | 5.680 (3.006) [a] | 0.1868 (2.028) [b] | 0.6473 (5.540) [a] | 0.833 | 482.94 | 0.9281 |

[a,b,c] denotes significance at 1%, 5%, and 10%; [1] LL: log likelihood value.

In the GARCH (1, 1) model, the sum of $\alpha_i + \beta_i$ acts as a response function of the persistence of volatility. Volatility persistence measures the extent to which the shocks within the index remain persistent. In the full period, the persistence of the shock to volatility is found to be stable, except for MENA, Qatar, Bahrain, and UAE. The volatility shock tends to remain persistent for a longer period in ASEAN, Australia, BRIC, GCC, and MENASA Islamic stock indexes as the persistence measure is not significantly different from unity. For instance, in the case of the MENA Islamic stock exchange, the volatility response function tends to decay quickly after 30 days ($0.749^{30} = 0.000171$) as compared to the Australian index ($0.971^{30} = 0.414$). Since the response function to volatility will define the persistence of the shock and relative riskier financial series, the closer the value of $\alpha_i + \beta_i$ to 1, the longer the shock will remain in the system.

Comparing the volatility response function among Islamic stock indices in pre- and post-COVID-19 periods reveals that the shocks in the pre-crisis period would likely decline in a shorter period. For the MENA Islamic stock index, after one month, the proportion of the shock remains at 0.0358 ($0.895^{30}$), which is the highest shock persistence function in pre-crisis periods. However, the Qatar Islamic index is observed to have the weakest persistence factor before the outbreak of COVID-19, i.e., $0.522^{30} = 0.0000000033$. Despite this, the volatility response function has significantly increased from the pre- to post-crisis period in all selected Islamic indices, except for Qatar. The stock return volatility of the Qatar Islamic index is the least and was less affected by this COVID-19 crisis. With the exception of the Qatar Islamic index, our results are consistent with the previous studies, which implied that COVID-19 triggered a substantial increase in stock market volatility (Bahloul and Khemakhem 2021; Corbet et al. 2021; Engelhardt et al. 2021; Fakhfekh et al. 2021; Guru and Das 2021; Jeribi and Manzli 2021; Nguyen et al. 2021). However, the volatility persistence significantly increased from 0.699 to 0.958 for the ASEAN Islamic index, 0.812 to 0.961 for Australian Islamic index, 0.889 to 0.937 for Bahrain, 0.675 to 0.979 in the BRIC Islamic index, 0.776 to 0.957 in GCC, and 0.895 to 0.966 for the MENA Islamic index. The results imply that the volatility shocks in the Islamic stock indexes would prevail for a longer period after COVID-19, making Islamic indexes riskier after COVID-19 (Bahloul and Khemakhem 2021) compared to the pre-COVID-19 era.

## 5. Conclusions

This study aimed at examining how COVID-19 affected Islamic stock markets around the globe. Using the Thomson Reuters' Islamic stock indexes of Australia, Qatar, Bahrain, UAE, MENA, MENASA, BRIC, and GCC, this study captured the responses of all selected indices in short- and long-term event windows. To analyze if Islamic stock markets are immune to the recent pandemic and how its volatility changes, an event study methodology and GARCH (1, 1) model was employed in this study.

The results suggest that in the short run the Islamic stock markets' price movements diverged throughout the period under consideration. However, the Islamic stock indexes of Australia, UAE, MENASA, and Bahrain remained stable in the first 15 days of the declaration of COVID-19 as a pandemic. Bahrain and Australia were least affected by the crisis until mid-February 2020. However, investors from GCC, MENA, BRIC, Qatar, and ASEAN reacted in a rather pessimistic manner after initial pandemic news. The market showed a significant downturn even in the short-term window, though the CAR of GCC index is negative but insignificant. It is also evident that the spread of virus was not abrupt among all the countries until mid-February 2020. The news tends to play its role in driving investor sentiments across the globe, which leads to a significant downturn in the market index. The research further explores the long-term effect of this pandemic using the long-term event window (16, 43). According to the WHO, the virus reached 200 countries by the end of March 2020. Therefore, it is evident that all the Islamic stock indexes were significantly affected in the long-term window by the crisis. At the same time, the Islamic stock index of Bahrain regained a positively significant cumulative return in the long-term window. The suspected reason behind this recovery is the least number of confirmed COVID-19 cases until 31 March 2020.

Together with our findings, the volatility persistence measure showed a significant increase in the volatility of all selected Islamic stock indices except Qatar, Qatar being the least affected region in terms of an adverse stock market reaction after COVID-19. Additionally, from the time series of the Qatar Islamic index, it is evident that Qatar was among the least traded stock index compared to other Islamic indices. Notably, the BRIC Islamic index suffered the highest degree of volatility persistence (45%) in post-crisis periods. MENASA being the least affected in terms of volatility persistence. Therefore, the markets were drastically affected by COVID-19 around the globe. Finally, it is concluded that due to the severity of the pandemic, the economic activities were affected badly due to the lockdown situation. Consequently, the natural ability of Islamic stocks to absorb external shocks suffered a lot due to limitations imposed due to lockdowns in every country.

The results of this research have implications for policymakers and investors in two ways. Firstly, policymakers should have been signaling their confidence more in the economy turning back to its normal path. Additionally, the role of media and the news agencies should be under strict compliance by the government to avoid panic among investors. Media could have played a significant role in controlling panic across countries. Secondly, investors should be more vigilant and need to spread their risk in a healthy portfolio to deal with such uncertainties.

Our study provides a significant contribution concerning the nature of Islamic stock markets. However, this study has a few limitations due to the availability and lack of data. For instance, market capitalization of Islamic stock markets could not be analyzed, which could provide better insights into investor's behavior in response to the global health emergency. However, there is considerable room for future research to study investors' sentiments with respect to Islamic stock markets. Importantly, the research can be refined in future by comparing the Islamic and conventional stock markets in response to external shocks, i.e., health or economic crises. The impact of the financial crisis of 2007–2009 and COVID-19 could provide a fruitful direction for further research in this field, subject to the availability of data.

**Author Contributions:** A.S. and J.S. worked on the conceptualization and research design, investigation, data analysis, software, and methodology. J.B. contributed to supervision. However, J.S. contributed in reviewing, supervising, and editing the paper. All authors have read and agreed to the published version of the manuscript.

**Funding:** This research received no external funding.

**Institutional Review Board Statement:** Not applicable.

**Informed Consent Statement:** Not applicable.

**Data Availability Statement:** Data are sourced from Thomson Reuters data stream, which is available on demand.

**Acknowledgments:** The authors would like to thank the Hungarian University of Agriculture and Life Sciences, Hungary, Budapest Business School, and Tempus Public Foundation for their support. Furthermore, the authors would like to thank and acknowledge the efforts and time given by Kiss Gábor Dávid (Associate Professor—University of Szeged) and Pejman Ebrahimi (Ph.D. student at MATE, Hungary) for improving the quality of this paper.

**Conflicts of Interest:** The authors declare no conflict of interest.

## Appendix A

**Table A1.** Daily cumulative average abnormal return across all indices.

| Days | CAAR$i$ | Standard Error | $t$ Test | $p$ Value |
|---|---|---|---|---|
| 0 | 0.0034 * | 0.0017 | 1.9405 | 0.0682 |
| 1 | 0.0014 | 0.0017 | 0.7945 | 0.2748 |
| 2 | −0.0068 * | 0.0021 | −3.2083 | 0.0094 |
| 3 | −0.0036 * | 0.0017 | −2.1772 | 0.0476 |
| 4 | −0.0014 | 0.0013 | −1.1418 | 0.1960 |
| 5 | −0.0035 * | 0.0015 | −2.3790 | 0.0348 |
| 6 | −0.0076 * | 0.0014 | −5.5476 | 0.0003 |
| 7 | −0.0115 * | 0.0013 | −9.1186 | 0.0000 |
| 8 | −0.0115 * | 0.0013 | −9.0715 | 0.0000 |
| 9 | −0.0146 * | 0.0026 | −5.6537 | 0.0003 |
| 10 | −0.0165 * | 0.0018 | −9.0944 | 0.0000 |
| 11 | −0.0193 * | 0.0020 | −9.8770 | 0.0000 |
| 12 | −0.0192 * | 0.0015 | −13.246 | 0.0000 |
| 13 | −0.0214 * | 0.0020 | −10.8505 | 0.0000 |
| 14 | −0.0170 * | 0.0021 | −8.2094 | 0.0000 |
| 15 | −0.0129 * | 0.0054 | −2.4151 | 0.0329 |
| 16 | −0.0151 * | 0.0012 | −12.9465 | 0.0000 |
| 17 | −0.0266 * | 0.0025 | −10.5639 | 0.0000 |
| 18 | −0.0272 * | 0.0022 | −12.4954 | 0.0000 |
| 19 | −0.0395 * | 0.0032 | −12.2235 | 0.0000 |
| 20 | −0.0350 * | 0.0016 | −22.1255 | 0.0000 |
| 21 | −0.0546 * | 0.0058 | −9.4862 | 0.0000 |
| 22 | −0.0681 * | 0.0048 | −14.1733 | 0.0000 |
| 23 | −0.0588 * | 0.0028 | −20.6506 | 0.0000 |
| 24 | −0.0728 * | 0.0039 | −18.7999 | 0.0000 |
| 25 | −0.0690 * | 0.0021 | −32.5069 | 0.0000 |
| 26 | −0.0909 * | 0.0068 | −13.3106 | 0.0000 |
| 27 | −0.1526 * | 0.0098 | −15.5228 | 0.0000 |
| 28 | −0.1364 * | 0.0059 | −23.1252 | 0.0000 |
| 29 | −0.1301 * | 0.0052 | −24.8228 | 0.0000 |
| 30 | −0.1676 * | 0.0061 | −27.648 | 0.0000 |

**Table A1.** *Cont.*

| Days | CAAR*i* | Standard Error | *t* Test | *p* Value |
|---|---|---|---|---|
| 31 | −0.1746 * | 0.0073 | −23.8701 | 0.0000 |
| 32 | −0.2041 * | 0.0116 | −17.6101 | 0.0000 |
| 33 | −0.2215 * | 0.0082 | −27.1397 | 0.0000 |
| 34 | −0.2270 * | 0.0107 | −21.1689 | 0.0000 |
| 35 | −0.2387 * | 0.0086 | −27.6477 | 0.0000 |
| 36 | −0.2163 * | 0.0091 | −23.6746 | 0.0000 |
| 37 | −0.2602 * | 0.0121 | −21.4901 | 0.0000 |
| 38 | −0.2558 * | 0.0049 | −52.7113 | 0.0000 |
| 39 | −0.2286 * | 0.0053 | −43.4634 | 0.0000 |
| 40 | −0.2288 * | 0.0065 | −35.4653 | 0.0000 |
| 41 | −0.2291 * | 0.0076 | −30.3157 | 0.0000 |
| 42 | −0.2444 * | 0.0093 | −26.3424 | 0.0000 |
| 43 | −0.2337 * | 0.0059 | −39.9243 | 0.00000 |

Note: * significant at 5%.

**Appendix B**

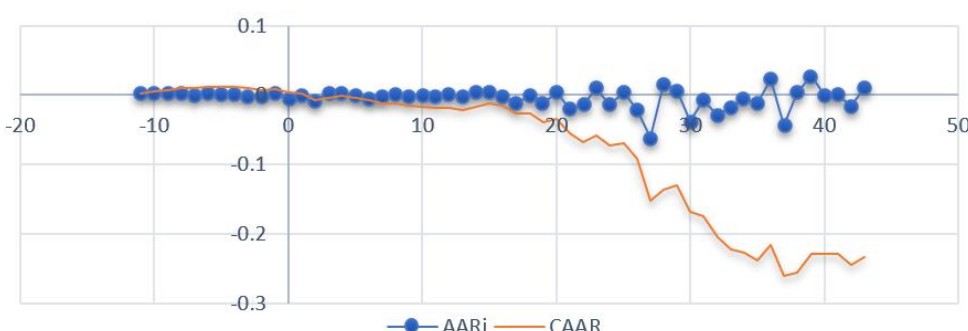

**Figure A1.** Cumulative abnormal return (CAR) of Thomson Reuters Islamic stock indices.

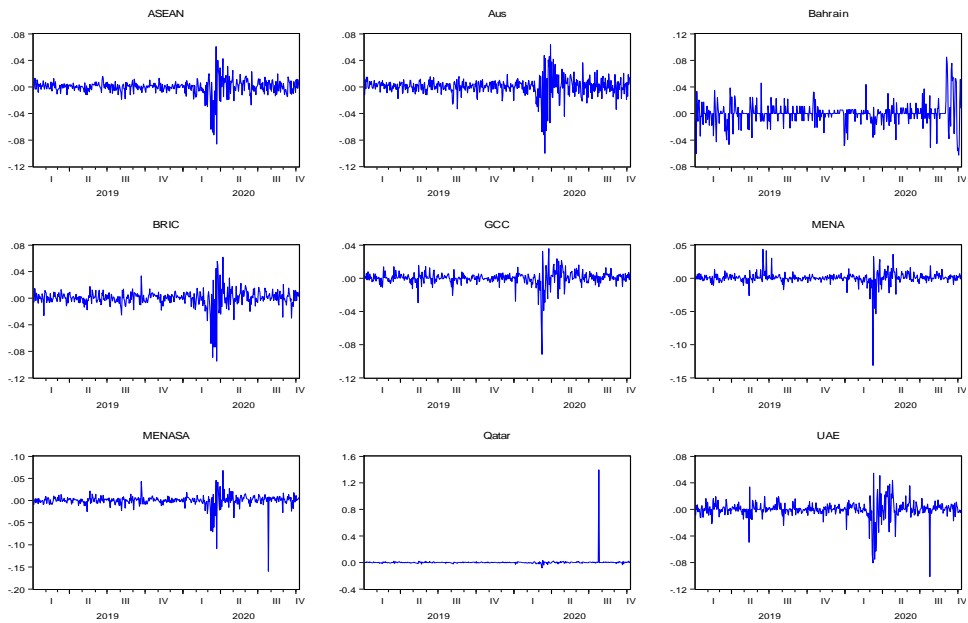

**Figure A2.** Full period. Source: authors' calculation using EViews 10.

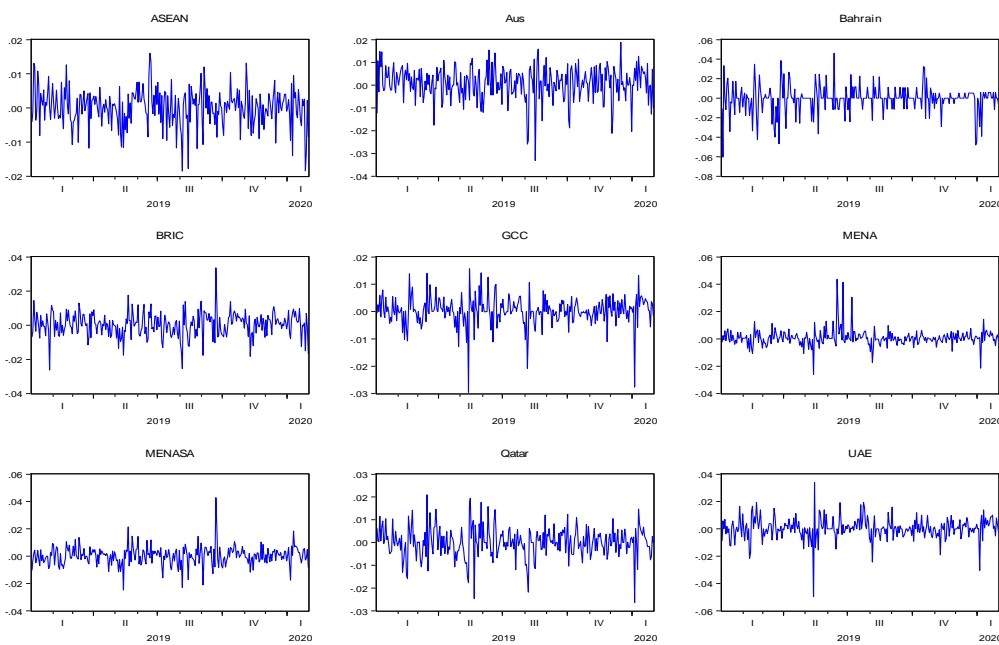

**Figure A3.** Pre-Crisis Period. Source: Authors' calculation using EViews 10.

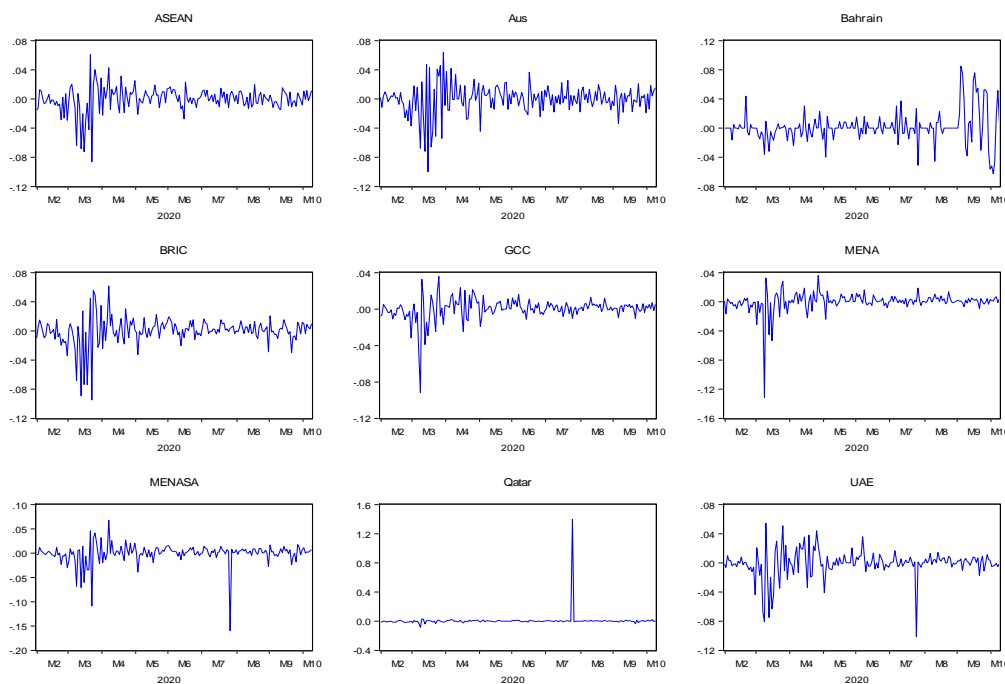

**Figure A4.** Post-crisis. Source: authors' calculation using EViews 10.

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
