# Peer review of "COVID-19 and Islamic Stock Index: Evidence of Market Behavior and Volatility Persistence"

_jrfm, doi:10.3390/jrfm14080389_

Round 1

Reviewer 1 Report

Dear Authors, 

Thank you for sending the revised version of your paper. While the paper have been improved, It looks that the introduction and the literature review should be clearly separated. Similarly, the contribution of the study should be stated in the introduction. 

Also, suggestion for future research and limitation is needed in the conclusion. 

The abstract should be rewritten to reflect the research methodology. 

Regards, 

Reviewer

Author Response

Thank you for sending the revised version of your paper. While the paper has been improved, It looks that the introduction and the literature review should be clearly separated. Similarly, the contribution of the study should be stated in the introduction. 

Response: Thank you for suggesting to separate the introduction and literature review sections. We have changed the structure according to the reviewer’s comment. Furthermore, we mentioned the contribution and purpose of the study at the end of introduction section (lines 88-99).

Also, suggestion for future research and limitation is needed in the conclusion. 

Response: Thank you for your suggestion, we have now added the limitation and future research to the end of the conclusion part, in lines 671-680.

The abstract should be rewritten to reflect the research methodology. 

Response: We have agreed with the reviewer and rewrote part of the abstract in order to reflect the purpose and research methodology of the study (lines 9-28).  

For the English proofreading, we asked for a native English professional at this phase as well.

We hope our revision meets your expectations.

Reviewer 2 Report

The study uses the event study methodology to analyze the market reaction to Covid-19 announcements.

The main problem I see is that the authors claim to have used the market model for this analysis. This means that all their results rely on the extent to which the indices chosen for their analysis reacted additionally to how the market reacted. Abnormal returns are computed as differences between realized values and predicted values according to the market. This is why their Figure 1 shows very random reactions in the immediate post-event window. All markets reacted significantly in the same manner so a market model used for an event study will only reveal what kind of additional reaction (if any) was recorded by the particular indices as compared with the index chosen as market (in the market model).

If the analysis shifts towards the identification of how particular indices reacted (outbreak in Australia as opposed to global market or outbreak in Bahrain as opposed to the global market) then we could have a valid point. This means that in the market model, a global index should be chosen.

Author Response

The study uses the event study methodology to analyze the market reaction to Covid-19 announcements.

The main problem I see is that the authors claim to have used the market model for this analysis. This means that all their results rely on the extent to which the indices chosen for their analysis reacted additionally to how the market reacted. Abnormal returns are computed as differences between realized values and predicted values according to the market. This is why their Figure 1 shows very random reactions in the immediate post-event window. All markets reacted significantly in the same manner so a market model used for an event study will only reveal what kind of additional reaction (if any) was recorded by the particular indices as compared with the index chosen as market (in the market model).

If the analysis shifts towards the identification of how particular indices reacted (outbreak in Australia as opposed to global market or outbreak in Bahrain as opposed to the global market) then we could have a valid point. This means that in the market model, a global index should be chosen.

Response: Thank you for this valuable comment. We certainly agree that comparing the outbreak in one market as opposed to the market chosen for comparison is valid if the market reflection is chosen from global index. However, we have considered Thomson Reuter’s Global Index to reflect the market and to calculate the expected return (which further being used to find abnormal returns). Thomson Reuter issued the Islamic index of several countries and regions in April 2009. The Thomson Reuter’s Islamic global index is composed of 20 countries and 3 regions to reflect the Islamic market globally. Hence, in our study we used 9 different indices including Islamic index of Australia, Qatar, UAE, Bahrain, GCC, MENA, MENASA, BRIC, and ASEAN compared their actual return from the market i.e. Thomson Reuter’s Global Islamic Index.

Round 2

Reviewer 1 Report

Dear Authors, 

I think the paper has been greatly improved. However, the issue of persistent remains a challenge. This is because pandemic is not up to 2 years and long run is not possible. To address this setback, there is need to implement a Panel vector Autoregressive Model. This will provide more understanding on the prior behaviour of the series and show the robustness of the short and long run dynamics.  

Hope this help. 

Regards, 

Reviewer

Reviewer 2 Report

I think the interpretation should reflect the elements mentioned in the comments to reviewer. Additionally, the paper should also mention how the GARCH model was used. Besides the fact that many elements from the section that describes the use of GARCH model are too simple for a scientific paper (they are manual level), a clear mentioning of how the model was used to compute AR, CAR or for any other result of this paper would be important for the reader.

Round 3

Reviewer 1 Report

Dear Authors, 

I think the paper has been improved and can be published after minor language checks.

Regards, 

Reviewer

This manuscript is a resubmission of an earlier submission. The following is a list of the peer review reports and author responses from that submission.

Round 1

Reviewer 1 Report

The paper focuses on an interesting topic, with the authors studying the impact of COVID-19 on the returns and the volatility of 9 Islamic stock market indexes. Though it does not provide a major contribution to the literature, there is some novelty in this work. Overall, the paper is well organized, and the conclusions are clear.

The methodology is adequate, although it could be improved. More precisely, the authors state that “To estimate the significance of abnormal return on and after the event date we used t-test in this study” (line 227). To increase the robustness of the inferences, I suggest using also a non-parametric test. Besides, in line 466, the authors mention that “The GARCH (1,1) model is selected based on lowest AIC (Akaike Info criteria), maximum log likelihood and a higher value of adjusted R squared”. However, what were the alternatives considered? Asymmetric GARCH models?

I also suggest rewriting the first part of the “Abstract” (lines 10-15) to make it be more concise. Furthermore, the “Introduction” section is missing a final paragraph, containing the structure of the paper.

Finally, the paper needs another round of editing/proofreading in order to improve the quality of the writing, because many grammatical errors persist.

Reviewer 2 Report

The paper creates an event study to investigate the reaction of Islamic stock markets to the current pandemic. I think the paper would benefit a lot from a better explanation of why such a study is necessary, given the fact that, as the authors mention in the literature review section, there are other studies that established that the market showed a reaction. Is there a particular reason for which we should consider that Islamic markets should not react?

There is need for a serious review of English. Besides some situations with lack of coherence, like

  • in the abstract: "...authors utilized Using Event Study Methodology..."
  • rows 45 - 46: "...millions of people 45 affected badly affected with it..."

there are also several expressions that need reconsidering. Sometimes phrases have no meaning: 

"Since volatility clustering is evident from figure B-2, B-3, and B-4 (Appendix B).", row 447 - 448.

On the methodology side, fitting GARCH models before and after the event is not sufficient for a measure of change in volatility without clear tests of significance for the DIFFERENCE in the fitted coefficients. Furthermore, an event study in volatility is usually performed in a different manner in the literature, it follows closely the classical event-study methodology by comparing the forecasts from a GARCH model to the squared returns within a window around the event.

I think the paper needs a serious revision before being considered for publication.

Reviewer 3 Report

The author has examined the potential source of volatility using the GARCH model. The author highlighted the COVID events and some economic implication in short and longer term. The following are the flaws that make it impossible to recommend the study.

  • The methodology is not sufficient. GARCH and ARCH are not enough to justify the conclusion of the study. This is due to the fact that the procedure has significant flaws, and that convergence is impossible given the sample size.

  • A lot of repetition of ideas. 31 to 33 is the same with 64 to 65 and 149
  • Mixture of capital letter with small throughout the work.
  • The author is unable to make case for the study.
  • Line 57 to 58, According to the author, pandemics and exogenous shocks have a negative impact on the stock market, which has been reported in the literature. Then the important question is why is the study necessary?
  • Since evidence by Fakhfekh Jeribi and Ben Salem, 2021 and that of Insaidoo et al. (2021) reported negative, what else are the author providing to justify the novelty of their work.
  • The research question is not appropriate.

I'm not sure I can recommend think paper for publication because of the identified problem.